# Embedded Field Stalk Detection Algorithm for Digging–Pulling Cassava Harvester Intelligent Clamping and Pulling Device

**Wang Yang \*, Junhui Xi** **, Zhihao Wang, Zhiheng Lu, Xian Zheng, Debang Zhang and Yu Huang**

College of Mechanical Engineering, Guangxi University, Nanning 530004, China; junhui_xi@163.com (J.X.); wzh0186@163.com (Z.W.); 8812316@163.com (Z.L.); xian_zheng@163.com (X.Z.); zdbang1233@163.com (D.Z.); huangyu202305@163.com (Y.H.)
**\*** Correspondence: yanghope@163.com

**Abstract:** Cassava (*Manihot esculenta* Crantz) is a major tuber crop worldwide, but its mechanized harvesting is inefficient. The digging–pulling cassava harvester is the primary development direction of the cassava harvester. However, the harvester clamping–pulling mechanism cannot automatically adjust its position relative to the stalks in forward movement, which results in clamping stalks with a large off-center distance difficulty, causing large harvest losses. Thus, solving the device's clamping location problem is the key to loss reduction in the harvester. To this end, this paper proposes a real-time detection method for field stalks based on YOLOv4. First, K-means clustering is applied to improve the consistency of cassava stalk detection boxes. Next, the improved YOLOv4 network's backbone is replaced with MobileNetV2 + CA, resulting in the KMC-YOLO network. Then, the proposed model's validity is demonstrated using ablation studies and comparison tests. Finally, the improved network is embedded into the NVIDIA Jetson AGX Xavier, and the model is accelerated using TensorRT, before conducting field trials. The results indicate that the KMC-YOLO achieves average precision (AP) values of 98.2%, with detection speeds of 33.6 fps. The model size is reduced by 53.08% compared with the original YOLOv4 model. The detection speed after TensorRT acceleration is 39.3 fps, which is 83.64% faster than before acceleration. Field experiments show that the embedded model detects more than 95% of the time at all three harvest illumination levels. This research contributes significantly to the development of cassava harvesters with intelligent harvesting operations.

**Keywords:** digging–pulling cassava harvester; intelligent clamping; stalk section detection; YOLOv4 optimization algorithm; embedded platform

## 1. Introduction

Cassava (*Manihot esculenta* Crantz) is one of the world's three major crops, the third largest food crop in the tropical region, the world's sixth largest food crop, and 1 billion people's food rations, known as "the king of starch". It is widely planted in more than 100 countries or regions in Asia, Africa, and Latin America and also an important energy crop and industrial raw material [1–3]. However, cassava harvesting primarily relies on manual methods, where farmers either pull up the tubers by hand or use a type of manual lever harvester [4]. This heavy reliance on manual labor for harvesting severely restricts the advancement of the cassava industry. Consequently, there is a critical need for research and development in cassava harvesting machinery.

Currently, three types of cassava harvesting machines are available [5]. The first type is the digging type of cassava harvester, which includes the 4UMS390II cassava harvester designed by Xue et al. [6], the TEK cassava harvester in Ghana [7], and the cassava harvester model P900 in Brazil [8], as shown in Figure 1a. This type of harvester can only function at semi-mechanized operation and requires manually pulling out the tubers after loosening

the soil. Although they exhibit lower operational efficiency, they are well suited for various soil conditions. The second type is the digging and shaking separation type of cassava harvester, which includes the vibrating cassava root harvester developed by Gupta. et al. [9] in Thailand, the classic model API cassava harvester developed in Malaysia [10], the 4UMZ-1400 rear-collected cassava combine harvester designed by Li et al. [11], and the 4UM-160 cassava harvester developed by Mo and Huang [12], which adopts the technical route of shoveling (cutting), soil breaking, soil sieving, separating–lifting, and placing, as shown in Figure 1b. This kind of harvester can function at the fully mechanized level, with great operating efficiency, but it consumes a lot of power and is poorly adapted to diverse soil types where cassava is cultivated, with less tuber loss and damage in sandy soils and more tuber loss and damage in clayey soils. The third type is the digging–pulling cassava harvester, which includes the cassava root digging–pulling type of harvester in Cuba [5], the cassava harvester developed by the University of Leipzig in Germany [13], and a series of digging–pulling types of cassava harvesters developed by Hainan University [14] and Guangxi University [15], as shown in Figure 1c, among others. This kind of harvester can achieve fully mechanized operation, relatively high operational effectiveness, low power consumption, and excellent soil adaptability. Therefore, the primary direction of cassava harvester development has been the digging–pulling type.

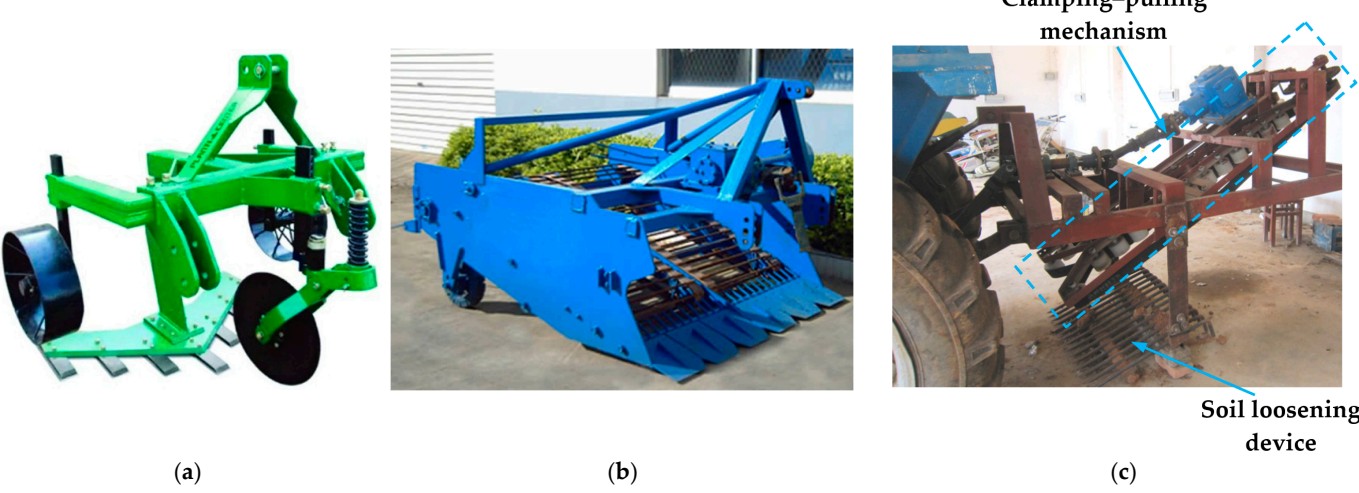

(**a**)　　　　　　　　　　　(**b**)　　　　　　　　　　　(**c**)

**Figure 1.** Typical models in three types of mainstream cassava harvesters: (**a**) digging cassava harvester; (**b**) digging and shaking separation type cassava harvester; (**c**) digging–pulling cassava harvester.

As illustrated in Figure 1c, the digging–pulling cassava harvester consists of a soil loosening device, a stalk clamping–pulling mechanism, and other components. The traditional working principle is as follows: during harvesting, the digging shovel loosens the soil first, and then, the clamping–pulling mechanism, located directly above it, clamps the stalks, pulls up the tubers, and conveys the tubers to the upper rear. Since cassava stalks present a thin and tall condition, the stalks tend to grow at an incline due to wind force, resulting in deviation from the vertical direction, as shown in Figure 2. The off-center distance is defined in this paper as the distance of the cross-section of cassava stalks 30 cm above the ground at harvest from the centerline between the two furrows, as illustrated in Figure 3. According to a significant number of field statistics, 84.5% of the stalks had an off-center distance of less than 200 mm, 9.3% had an off-center distance of 200–300 mm, and 6.2% had an off-center distance of more than 300 mm [16]. However, the existing digging–pulling harvester's clamping–pulling mechanism cannot automatically adjust its position relative to the stalks in forward movement, which results in clamping stalks with large off-center distance difficulty, causing large harvest losses. Thus, solving the device's clamping location problem is the key to loss reduction in the harvester. But the traditional

sensor fusion approach [17] is difficult to apply to stalk clamping with a considerable off-center distance. As a result, in this work, stalks are recognized during the harvesting stage by carrying an embedded detection platform to gather stalk position information relative to the harvester. These data will be utilized to direct the automatic adjustment of the movement of the clamping–pulling mechanism, which will improve stalk clamping success and reduce harvesting loss.

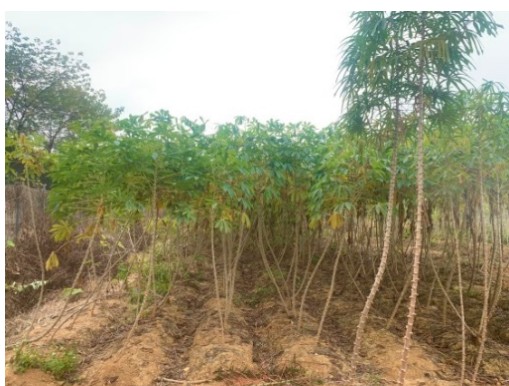

**Figure 2.** The growth of cassava stalks.

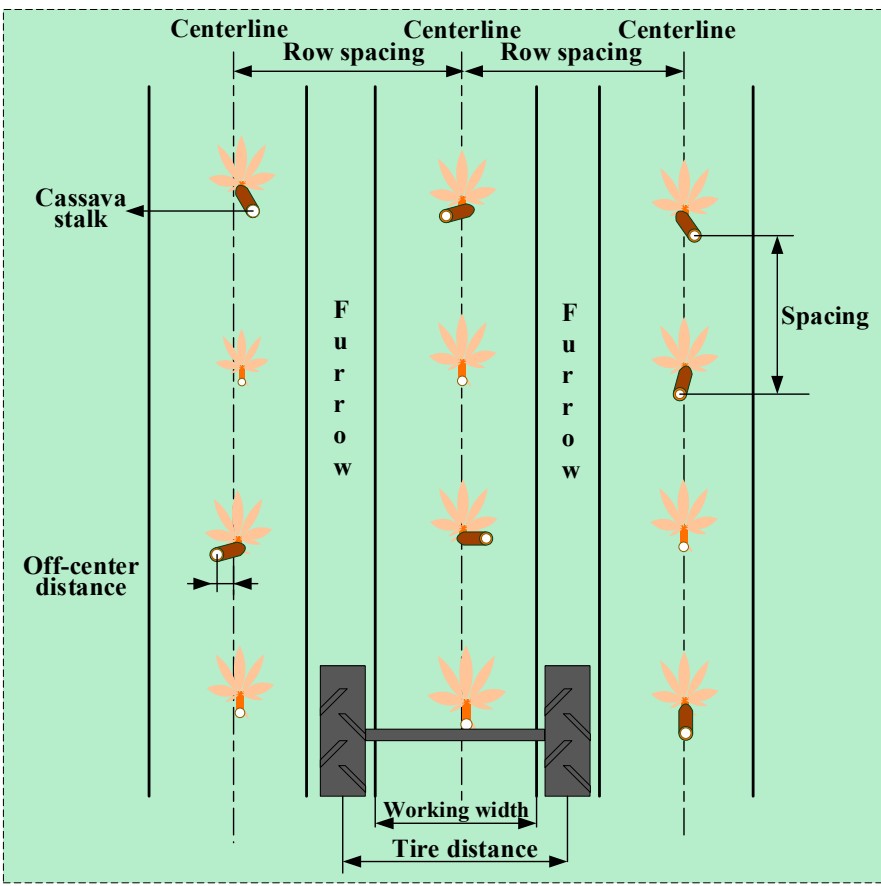

**Figure 3.** Cassava planting patterns and cassava stalk distribution during harvest.

Nowadays, object detection has become an effective means to assist in the positioning of mechanical structures, which can help the digging–pulling cassava harvester obtain an accurate position of the stalk for clamping. Different object detection algorithms have been successively applied to the detection and positioning of agricultural products by

researchers [18] and have achieved certain results. Among the two-stage detection algorithms, the representative Faster R-CNN algorithm has been used for fruit detection and has achieved great detection results [19]. Manual kiwifruit harvesting in orchards is labor-intensive, and Song et al. [20] have developed a VGG16-based Faster R-CNN all-weather working harvesting robotic vision system with good detection of kiwifruit images collected under different lighting conditions (morning, afternoon, and evening). A robotic vision system for multi-class fruit detection based on Faster R-CNN has been proposed by Wan and Goudos [21], who have established a library of outdoor fruit images and optimized the structure of the convolution and pooling layers in the model, and the mAP (mean Average Precision) of apples, mangoes, and oranges was above 91%. In the one-stage detection algorithm, Peng et al. [22] replaced the VGG16 network of the SSD algorithm with the ResNet-101 network for fruit detection, achieving an average accuracy of 88.4%. Gai et al. [23] improved the YOLOv4 model for cherry fruit detection by changing the prior bounding box to a circular marker box in line with the fruit shape and replacing the backbone network with the DenseNet network model, achieving a higher mAP value than YOLOv4.

Compared to the detection of fruit, the cassava stalk section is more difficult to distinguish from the soil due to the similar color, especially during field cassava tuber harvesting when illumination values vary greatly. Therefore, fully considering the influencing factors in constructing a cassava stalk dataset is crucial for successful field detection. Quan et al. [24] proposed an improved Faster R-CNN model based on a Field Robot Platform (FRP), which used five industrial USB cameras for data acquisition to capture a large number of sample images from different shooting angles. This system provides the basis for the extraction and detection of maize seedlings at different growth stages in a complex field environment. Junos et al. [25] used an improved YOLO model to detect scattered fruits in the oil palm plantation and adopted data enhancement methods such as fuzzy enhancement to simulate the actual natural environment. The results showed an outstanding average precision of 99.76% of the UAV image, with a detection time of 34.06 ms. Overall, object detection algorithms have shown great potential for improving agricultural automation, but their effectiveness depends on various factors such as lighting conditions, image quality, and dataset construction.

Although object detection algorithms are increasingly being utilized in agricultural engineering, the working environment of agricultural machinery in the field is terrible, and the expense of establishing large-scale computing platforms on it is prohibitively expensive, with no guarantee of dependability. Embedded platforms are tiny in size and can provide greater performance and reduced power consumption, as well as greater dependability and security and field edge computing at a cheaper cost [26]. On the other hand, the computational complexity of deep learning networks remains a challenge for their practical implementation in vehicular field environments with limited computing power. To address this issue, researchers have proposed various methods to simplify the current object detection models and improve real-time detection efficiency. Simplifying the network structure is an effective method to reduce the amount of calculation. For instance, Fu et al. [27] constructed the YOLO-Banana network by simplifying the network layer, which reduced the model size and shortened the detection time, providing a broad application prospect for the intelligent management and harvest in banana orchards. Another approach is to streamline the model by using a lightweight network structure to improve detection accuracy. The MobileNet lightweight network [28] with a streamlined architecture is one of the most successful application models. This network uses a depthwise separable convolution operation instead of standard convolution, which reduces the amount of calculation several times and greatly improves the network operation speed. On this basis, the MobileNetV2 lightweight network [29] uses an inverted residual, removing ReLU to avoid information loss. In addition, the MobileNetV3 lightweight network [30] uses neural architecture search (NAS) parameters to redesign the time-consuming layer structure and introduce the attention mechanism, making the network more accurate and efficient.

For existing object detection algorithms, researchers have also proposed corresponding lightweight versions, such as Light-Head R-CNN [31], Tiny-YOLO, Tiny-SSD [32], etc.

The objective of this study is to develop a real-time cassava stalk position detection method for adjusting the movement of the clamping–pulling mechanism. Firstly, the field environment and the growth state of the cassava are analyzed to determine a real-time cassava stalk coordinate acquisition scheme. Next, an image acquisition platform is established to acquire cassava stalk images and build a dataset. Then, the YOLOv4 model is adapted to improve and simplify its structure. The accuracy and real-time performance of the improved network are compared with those of different models to meet the detection requirements. Finally, the optimal detection model is deployed into the NVIDIA Jetson AGX Xavier (hereafter referred to as Xavier) embedded device for field trial validation. The results of this paper can provide important technical support for the development of digging–pulling cassava harvesters and intelligent harvesting research.

## 2. Materials and Methods

### 2.1. Dataset Production

#### 2.1.1. Image Acquisition

The outdoor image acquisition site is located in cassava cultivation area A, at the teaching and research base of Guangxi University College of Agriculture, Nanning, Guangxi, China (22°51′ N, 108°17′ E). Cassava was cultivated in a conventional ridged pattern with row spacing of 1 m and plant spacing of 0.7–1 m. As shown in Figure 3, the stalks grew to a height of 2–2.5 m and their diameters ranged from 25–40 mm. Like with manual cassava harvesting, stalks more than 30 cm above the ground must be manually chopped off and debris swept away before the digging–pulling cassava harvester operation. As displayed in Figure 4, the stalks that remained after truncation varied in thickness and attitude, and there were various off-center positions in the cassava stalks.

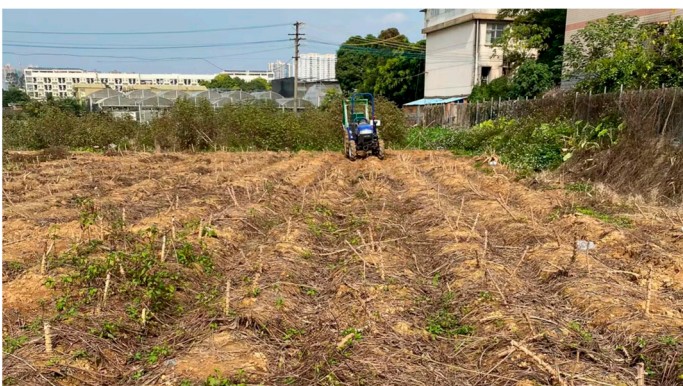

**Figure 4.** Situation of cassava stalks cut off before harvesting.

The field movement of harvesters is complicated. In order to collect stalk images while the harvester is moving in the field and reduce the influence of the cassava stalk postures and the complexity of detection, a field acquisition platform was constructed on a digging–pulling cassava harvester developed by Guangxi University, as shown in Figure 5. The Xavier development board (NVIDIA Corporation, Santa Clara, CA, USA), camera and lens, and related supporting equipment (as shown in Table 1) were installed on the harvester. Specifically, the industrial camera was mounted on the front end of the cassava harvester frame, and the lens was arranged vertically in line with the field to detect the top section of the cassava stalk, as shown in Figure 6. The camera was arranged before the clamping mechanism to allow enough time for the movement of the clamping–pulling mechanism. The camera was arranged vertically in line with the field so that the camera field of view and the lower plane of the frame can be roughly coincident, and a machine coordinate system was constructed in which the camera detection field of view and the frame of the harvester coincide. In this way, the harvester can more easily obtain the

position information of the cassava stalks relative to the rack, facilitating the harvester clamping device to directly obtain the position coordinates of the cassava stalks for real-time adjustment and also reducing the use of arithmetic power. During the acquisition process, the cassava harvester simulates the harvesting state by driving along the ridge in a straight line. When the collector observes that the cassava stalk cross-section is fully presented on the screen, the cassava stalk cross-section photographs are collected manually.

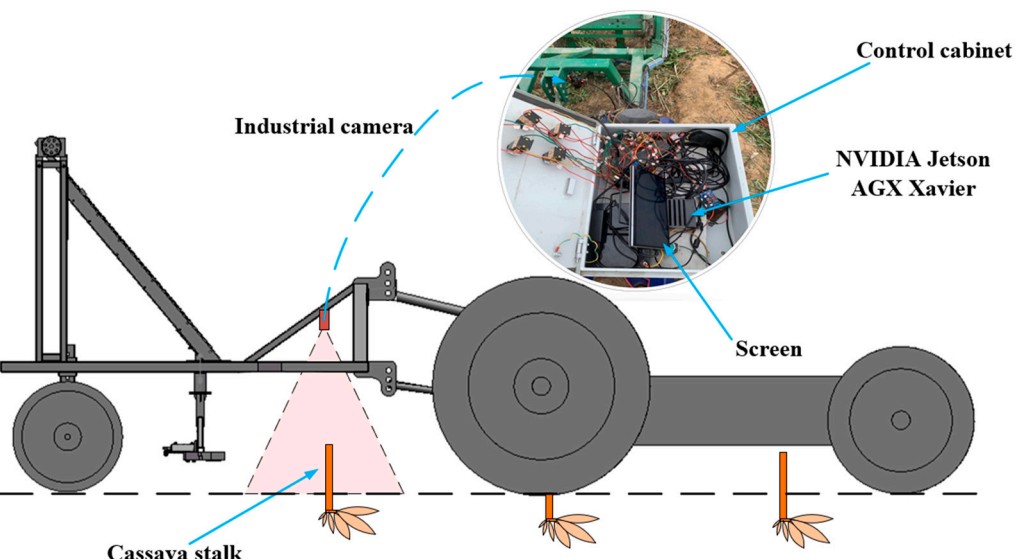

**Figure 5.** Field image acquisition platform.

**Table 1.** Image acquisition platform equipment.

| Carrier | Computing Platform | Camera/Lens |
|---|---|---|
| Lovol M504-E tractor | NVIDIA Jetson AGX Xavier | Alvium 1800 U-508M/ KOWA LM8JC5MC |

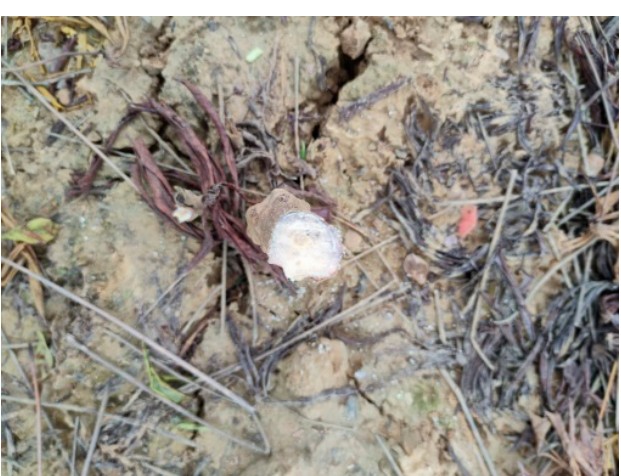

**Figure 6.** Cassava stalk truncated cross-section (test section).

The working principle of a digging–pulling cassava harvester equipped with an embedded inspection platform is as follows: when the cassava harvester travels along the ridge in a straight line, the camera, vertically arranged at the front end of the cassava harvester frame, acquires the positional information about the cross-section of the cassava stalks on the ridge in the field of view and transmits the information to the slave computer (Programmable Logic Controller) after processing. The slave computer controls the

movement of the clamping–pulling mechanism according to the information, and finally, the clamping–pulling mechanism accurately clamps the cassava stalks and completes the pulling and harvesting action.

In this study, images were captured using OpenCV paired with Vimba5.0, which is the official development package for Allied Vision USB3.0 interface cameras (Allied Vision Technologies Gmbh., Stuttgart, Germany). The field of view for object detection is approximately 600 mm × 500 mm, with a detection height of 600 mm. The camera has $5 \times 10^6$ pixels with a resolution of 0.25 mm per pixel. However, changes in illumination intensity during harvest can lead to changes in the color presented by the object, resulting in large differences in chromaticity information. The illumination intensity under natural conditions, such as cloudy days and evenings, ranges from 500 lux to 6000 lux, and the soil will behave in a non-granular manner. On a sunny day, when the illumination intensity is greater than or equal to 100,000 lux, the cassava stems will appear as white reflected light [33]. Therefore, to represent these variations in illumination intensity, an illumination intensity detector (Dongguan Wanchuang Electronic Products Co., Ltd., Dongguan, China, range 1–200,000 Lux, measurement accuracy (±4% rdg ± 10 dgt for ≤10,000 Lux, ±5% rdg ± 10 dgt for >20,000 Lux)) was used to detect the illumination intensity before capturing the image, as shown in Figure 7a. Then, the images can be classified accurately.

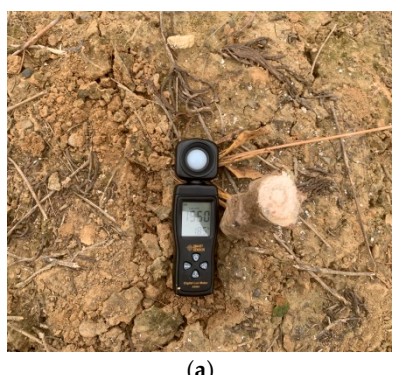
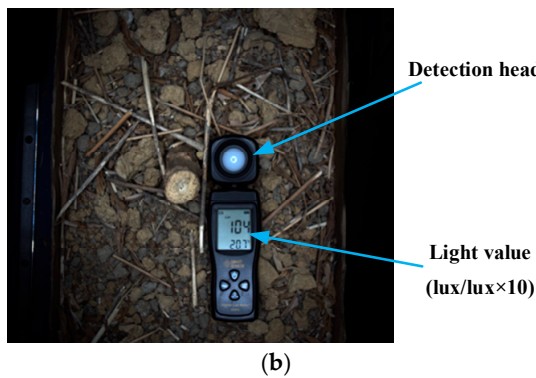

Detection head

Light value
(lux/lux×10)

(**a**)          (**b**)

**Figure 7.** Detection of light intensity: (**a**) field; (**b**) indoor.

When capturing images of cassava stalk sections in the field, it is necessary not only to cut off the stalks as required but also to consider the range of illumination values for image acquisition. The crop harvest is time-sensitive and cannot be put on hold due to weather or light, and the truncated cassava stalks are very susceptible to mold after being wetted by rain. It is very difficult to acquire effective images in the field, and in order to have enough images for model training, an indoor field image dynamic simulation acquisition platform was built based on a simulated field acquisition environment, as shown in Figure 8, using a shade cloth with an adjustable LED light source to cooperate for shooting [34,35]. According to the range of illumination values, the illumination conditions of the indoor inspection platform were controlled by light intensity detection instruments as well as LED lighting devices, as shown in Figure 7b, to acquire image data under different illuminations. At the same time, the field motion condition was simulated with a panning device. A total of 1000 images were finally collected, of which 1/3 of the image data were acquired in the field and 2/3 of the cassava stalk section images were collected by the indoor acquisition platform. Finally, the illumination intensity was divided into weak light, normal light, and strong light according to the range of 500 lux to 6000 lux, 6000 lux to 100,000 lux, and greater than or equal to 100,000 lux, respectively, and the stalk side light images were additionally taken under random illumination intensity, for a total of four types of cassava stalk cross-section images, with the number of each type of image collection accounting for about one-fourth.

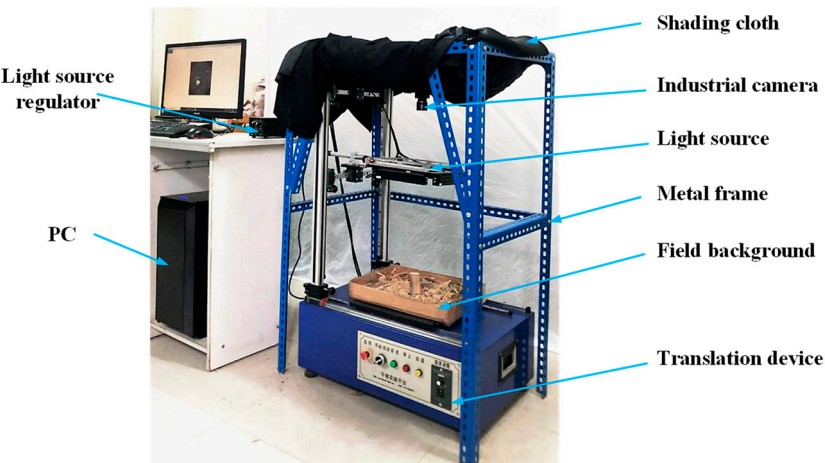

**Figure 8.** Indoor field image dynamic simulation acquisition platform.

### 2.1.2. Data Pre-Processing

This paper employed OpenCV coding to perform data augmentation on the original images so as to simulate the conditions that may be encountered in the field vehicle environment at the same time, and to improve model generalization and prevent the overfitting phenomenon. Prior to model training, the acquired raw images were randomly assigned in a 10:1 training-to-test set ratio. As illustrated in Figure 9, five data improvement methods, including motion blur, geometric changes, added noise, elastic changes, and random masking, were applied to the training and test sets, respectively. The image dataset was expanded to 11,000 pieces, comprising 10,000 images for the training set and 1000 images for the test set. As supervised learning in machine learning requires labeled datasets, GT labeling of cassava stalk cross-sections was performed using the image labeling tool LabelImg. The annotation results were stored in an XML file according to the PASCAL VOC format, which contains information such as target location, annotation box size, and category of the image, labeled as stem.

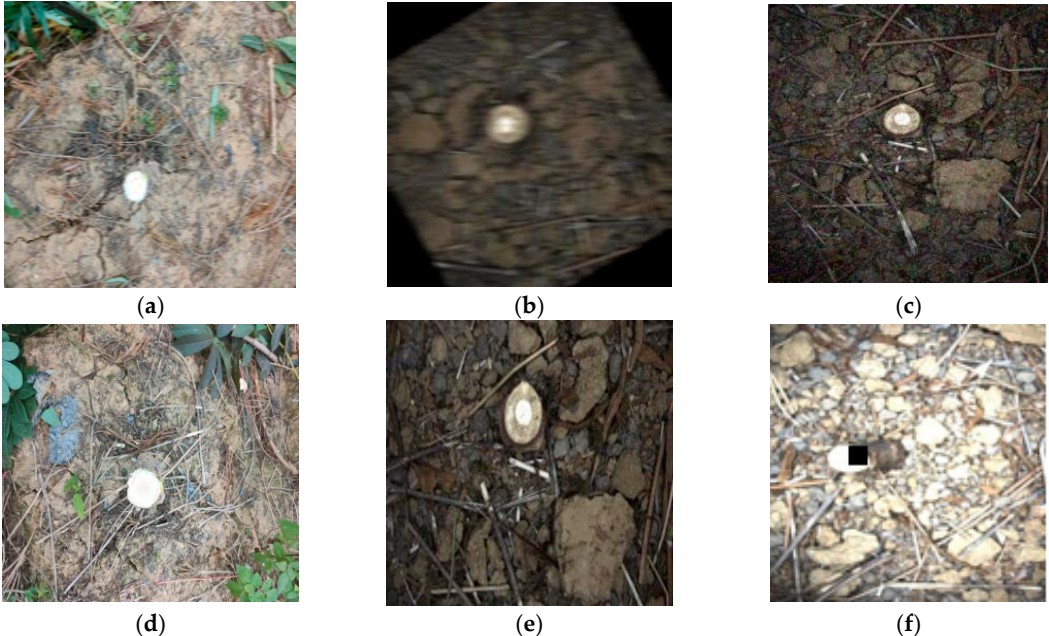

**Figure 9.** Data amplification method: (**a**) motion blur; (**b**) geometric rotation; (**c**) impulse noise; (**d**) Gaussian noise; (**e**) elastic transform; (**f**) random erasing (Black square is filled with zero pixel values).

*2.2. Classical Model*

2.2.1. YOLOv4

YOLOv4 is a typical one-stage detection algorithm that utilizes CSPDarknet-53 as the backbone network and the PANet (Path Aggregation Network) structure for multi-scale feature fusion. The network structure of YOLOv4, illustrated in Figure 6, comprises an input layer, a backbone network layer, a feature enhancement layer, and a classification regression layer. During the detection process, the image is partitioned into S×S grids, with each grid responsible for detecting the object whose center point is located within that region. Compared with the YOLOv3 network, Bochkovskiy et al. [36] incorporated the experience of CSPNet (Cross Stage Partial Network) and improved the backbone structure, which includes five CSP modules and $(3 + 2 \times X)$ convolutional layers in each CSPX, resulting in a total of 72 convolutional layers in the entire backbone. This modification addresses the issue of high forward computation in the network. Additionally, the Mish activation function [37] is used in the backbone, while the Leaky_ReLU activation function is still employed later in the network. The Mish function theoretically allows for a relatively small negative gradient inflow, thus ensuring information flow instead of an absolute zero bound as in ReLU. Moreover, the smoothed activation function enables better information penetration into the neural network, resulting in better accuracy and generalization.

2.2.2. Evaluation Metrics

For evaluating objective detection algorithms, performance evaluation metrics are necessary. The following two types of metrics are used to evaluate the models according to the evaluation metrics of neural network models.

(1)    Precision, Recall, and Average Precision (AP)

The confusion matrix in Table 2 is usually used in target detection to describe the detection.

**Table 2.** Confusion matrix.

| Confusion Matrix | | Predicted Position | Negative |
|---|---|---|---|
| Actual | Positive | True Positive (TP) | True Negative (TN) |
| | Negative | False Positive (FP) | False Negative (FN) |

Precision represents the ratio of the number of true positive samples to the total number of samples determined as positive, as shown in Equation (1). It measures the accuracy of positive predictions. Recall represents the ratio of the number of true positive samples to the number of all positive samples, as shown in Equation (2). It measures the ability of the model to retrieve all positive results. In order to balance between precision and recall, an evaluation index called average precision (AP) is commonly used to evaluate the effectiveness of object detection models. AP is calculated by computing the area under the precision–recall curve, as shown in Equation (3). In the context of cassava harvesting, the detection accuracy needs to be above 95% to ensure minimal harvesting losses and meet the requirements of the harvesting process.

$$Pre = \frac{TP}{TP + FP} \tag{1}$$

$$Recall = \frac{TP}{TP + FN} \tag{2}$$

$$AP = \int Precision \times Recall = \int_0^1 p(r)dr \tag{3}$$

(2)    Detection rate FPS

FPS (Frames Per Second) indicates the number of images detected in 1 s of time and is used to represent the detection rate. The object detection network should also ensure the speed of detection with high accuracy, and this system requires the detection model to detect at 30 fps or more on the embedded development board, thus realizing real-time object detection [38].

*2.3. Proposed Model*

The object detection network requires high accuracy for recognizing cassava stalk sections during harvesting in a vehicle motion state. Therefore, we optimized the YOLOv4 network, comparing it to other methods and adapting it to embedded devices. First, the cassava stalk section dataset produced in this paper was clustered and analyzed using the K-means algorithm, and the obtained anchor boxes were improved through multi-scale scaling to initially improve the detection accuracy of YOLOv4. The resulting network was named K-YOLO. Second, the network model was designed to be lightweight to create an efficient network computation model. Finally, an attention mechanism was added to guarantee network performance while ensuring a lightweight network design [39].

2.3.1. Anchor Box Clustering Analysis Based on the K-Means Algorithm

The anchor box is a rectangular frame based on the common size and proportion of the object to be detected, and its similarity to the prediction box is one of the important prerequisites for accurate object prediction. The core problem of the K-means clustering algorithm is how to represent the sample-to-sample distance, and different distance calculation methods yield different clustering effects. In this paper, we used the classic IOU (Intersection over Union) calculation method of the YOLO series to calculate the distance, as shown in Equation (4):

$$d(box, centroid) = 1 - IoU(box, centroid) \tag{4}$$

where *box* is the bounding box and *centroid* is the cluster center. The larger the *IOU* between *box* and the corresponding cluster center (anchor), the closer the distance.

2.3.2. Multi-Scale Scaling

The vertical distance during indoor shooting is relatively constant, and the aspect ratio and size of the stalk section are more consistent. The size of the anchor box obtained by distance-based clustering does not differ significantly across different sensing fields, which can impact the model's advantage of multi-scale output and does not align with the complex field conditions. Therefore, it is necessary to perform multi-scale random linear scaling on the obtained anchor box simultaneously in terms of aspect ratio. The scaling rules are as follows:

$$\begin{cases} x'_s = ax_s \\ x'_m = bx_m \\ x'_i = \frac{(x_i - x_s)}{(x_m - x_s)}(x'_m - x'_s) + x'_s \\ y'_i = x'_i \frac{y_i}{x_i} \end{cases} \tag{5}$$

where *a* and *b* denote the maximum and minimum scaling multipliers, $x_s$ and $x_m$ are the anchor frames before scaling, $x'_s$ and $x'_m$ denote the maximum and minimum anchor boxes, respectively.

2.3.3. Network Lightweighting

Using lightweight networks on embedded devices is a practical solution, and one such approach is to modify the K-YOLO network structure by adopting MobileNet. MobileNet is specifically designed for mobile or embedded devices. MobileNetV2 is an upgraded version of MobileNetV1, which introduces an image first before dimension raising and convolution, then the inverse residual structure of dimension reduction, and finally the

activation function of ReLU6. Moreover, it replaces the N × N matrix with the idea of 1 × N, N × 1, enabling the network to retain more low-dimensional information. MobileNetV3 further improves on MobileNetV2 by introducing the SE-Net (Squeeze and Excitation Network) attention module. However, SE-Net only considers measuring the importance of each channel by modeling channel relationships, ignoring the location information which is important for generating spatial coordinates. To solve this issue, the coordinate attention module (CA module) [40] was proposed, which considers not only the relationship between channels but also the location information in the feature space, thereby helping the model better localize and identify targets.

The K-YOLO network was improved by replacing the CSPDarknet53 backbone in the original structure with MobileNetV2 as the classification network, resulting in the KM-YOLO network. However, the streamlined model structure led to detection accuracy degradation. To solve this problem, the CA module was inserted between the global average pooling layer of the bottleneck and the 1 × 1 convolutional layer to obtain the KMC-YOLO network, as shown in Figure 10.

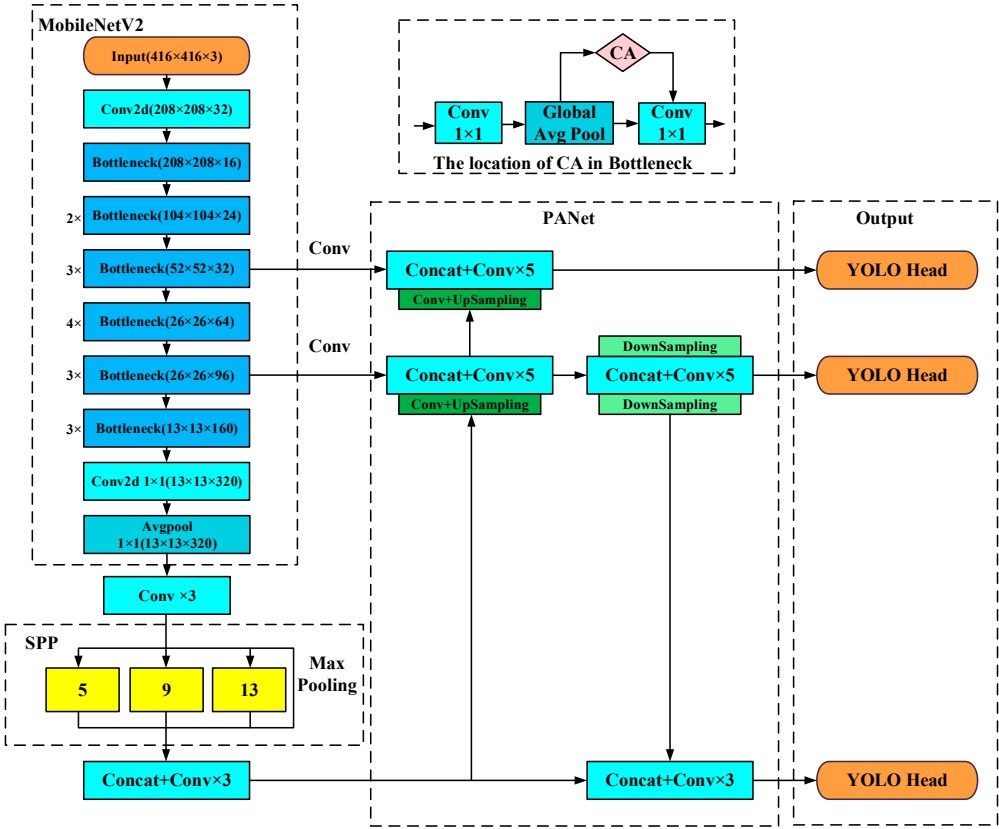

**Figure 10.** KMC-YOLO network structure: conv refers to convolution; SPP is spatial pyramid pooling; CA is coordinate attention; Conv is convolutional layer.

### 2.4. Training Platform and Environment

The object detection models were first trained and tested on a PC, using the PyTorch deep learning framework with CUDA and CUDNN installed for GPU acceleration. The training environment's composition and version are shown in Table 3.

**Table 3.** Algorithm operating environment.

| Name | Versions |
|---|---|
| Operating System | Ubuntu18.04 |
| CPU | Inter Core (TM) i5-7500 CPU @ 3.40 GHz, RAM:8.00 GB |
| GPU | NVIDIA GeForce GTX 2060-Super, RAM:8.00 GB |
| Compiler Environment | Pycharm |
| OpenCv | 4.1.0.25 |
| PyTorch | 1.7.0 |
| Python | 3.8 |
| CUDA, CUDNN | 11.0, 8.05 |

*2.5. Training Strategy*

2.5.1. Training Strategy for K-Means Clustering

To accelerate the model training and enhance its generalization ability, transfer learning [41,42] was employed in this study, whereby pre-trained weights of the COCO (Common Objects in Context) dataset were used as the backbone network during YOLOv4 training. The training was conducted for a total of 200 epochs. The initial learning rate was 0.0001; the learning rate at the end of training was 0.00001. The followings values were also obtained: a weight decay value of 0.0005, a warm-up parameter value of 2 epochs, and a batch size of 4. Optimization of the parameters was achieved using the small batch gradient descent method with Momentum = 0.9.

2.5.2. Training Strategy for Lightweight Networks

The KMC-YOLO network was a redesigned network structure without a corresponding pre-training model; thus, it was trained from scratch to update the weights without using transfer learning. During training, multi-scale training was enabled, warm-up was set to 2 epochs, and the weight decay value was adjusted to 0.0003 to accommodate the simpler network structure.

*2.6. Ablation Study and Comparison Test*

Ablation studies were built using the YOLOv4 model with the adoption of five network models to validate the efficacy of the suggested method. Instead of K-YOLO's backbone network, CSPDarknet53, the validation network in this research was developed using MobileNetV1, MobileNetV2, MobilnetV3, and MobileNetV3-small networks with the SE-Net attention module, respectively. The performance was then compared to that of the KMC-YOLO network.

Meanwhile, in order to provide a more comprehensive comparison, we conducted detection performance tests between our improved networks and existing mainstream networks and their improved versions. The compared models include YOLOv3-SPP, YOLOv4-tiny, YOLOX-tiny, YOLOv5s, and Faster R-CNN.

*2.7. Model Deployment*

The optimal network model was obtained after training and comparative validation of the network model on the PC side, but it needed to be deployed on the development board for practical applications. In this study, we used the Python environment manager Archiconda3_0.2.3 on Xavier for Python package installation and virtual environment creation [43]; the Xavier environment configuration is shown in Figure 11.

**Figure 11.** Development environment configuration.

The computing power of the embedded development board is limited, and running the trained deep learning model on a specific learning framework may lead to inefficient inference within the board. To overcome this issue, we used Xavier's TensorRT 8.0.1.6 to deploy the deep learning model on the development board. The implementation involved converting the PyTorch-trained model into the ONNX (Open Neural Network Exchange) intermediate format, which supports inference based on any framework, and then transforming it into the TensorRT engine based on the ONNX file format. Finally, efficient inference on the development board was achieved using the TensorRT engine file with FP16 floating-point precision.

## 3. Experiment Results and Discussion

### 3.1. Analysis of Training Results

#### 3.1.1. Representation of Multi-Scale Anchor Boxes

The cassava dataset was clustered with the "1-IOU" distance to obtain three sets of anchor boxes (IOU anchors), and the obtained IOU anchors were randomly scaled at multiple scales to obtain three new sets of anchor boxes (CIOU anchors), as shown in Table 4. The COCO anchors are the three sets of anchor boxes obtained from the original YOLOv4 model based on the COCO dataset.

**Table 4.** Anchor clustering results.

| Receptive Field | COCO Anchors | IOU Anchors | CIOU Anchors |
|---|---|---|---|
| Large | $(116 \times 90)$, $(156 \times 198)$, $(373 \times 326)$ | $(78 \times 77)$, $(72 \times 72)$, $(75 \times 61)$ | $(140 \times 114)$, $(124 \times 124)$, $(156 \times 154)$ |
| Medium | $(36 \times 75)$, $(76 \times 55)$, $(72 \times 146)$ | $(65 \times 67)$, $(62 \times 61)$, $(64 \times 50)$ | $(83 \times 65)$, $(72 \times 71)$, $(88 \times 91)$ |
| Small | $(12 \times 16)$, $(19 \times 36)$, $(40 \times 28)$ | $(53 \times 43)$, $(48 \times 50)$, $(55 \times 55)$ | $(26 \times 21)$, $(35 \times 48)$, $(36 \times 36)$ |

The model's detection performance was tested on a test set comprising 1000 images. Table 5 shows the mAP scores for the three sets of models. The results indicate that using K-means clustering to obtain IOU anchors for YOLOv4 improves the detection accuracy compared to using COCO anchors. Moreover, using multi-scale scaling to obtain CIOU anchors further improves detection accuracy. Among the models, the K-YOLO model using CIOU anchors achieved the highest mAP score of 95.6% in the test set. With GPU acceleration, the K-YOLO model using CIOU anchors achieved a detection speed of 19.23 fps.

**Table 5.** Model accuracy under different anchor boxes.

| Anchors | mAP (%) |
| --- | --- |
| COCO anchors | 92% |
| IOU anchors | 93.5% |
| CIOU anchors | 95.6% |

3.1.2. Representation of Improved Network

According to the training strategies, each of the two improved lightweight networks was trained for 200 epochs. The changes in loss values during the training of the KMC-YOLO network are shown in Figure 12.

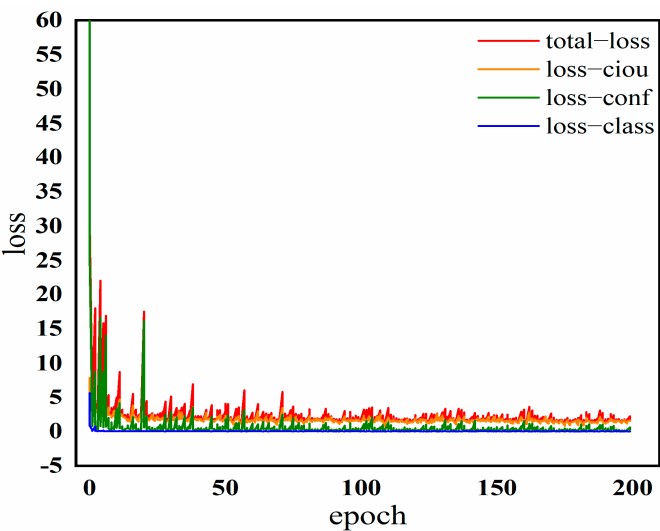

**Figure 12.** KMC-YOLO training process: the term total−loss refers to the overall loss, loss−ciou represents the CIoU loss, loss−conf represents the confidence loss, and loss−class represents the classification loss.

From Figure 12, it can be observed that the network loss values converge rapidly and decrease within 25 epochs. There are some small fluctuations in the confidence loss before 70 epochs, after which the fluctuations of each loss value become smaller and the convergence effect improves. The reduction in loss values for the newly designed model is quick and effective, indicating that the network structure design and hyperparameter settings are reasonable. Testing on a test set with 1000 images also shows that no overfitting occurs.

3.1.3. Validation of the Network Model

Analysis of KMC-YOLO Test Results

The test results of the KMC-YOLO network are further analyzed, and some of the results are shown in Figure 13. From the figure, it can be seen that the model demonstrates good detection performance on images captured under three different lighting conditions: bright, dark, and side light. The model also performs well on images with added processing effects such as motion blur, rotation, and distortion, including images with tilted stalks.

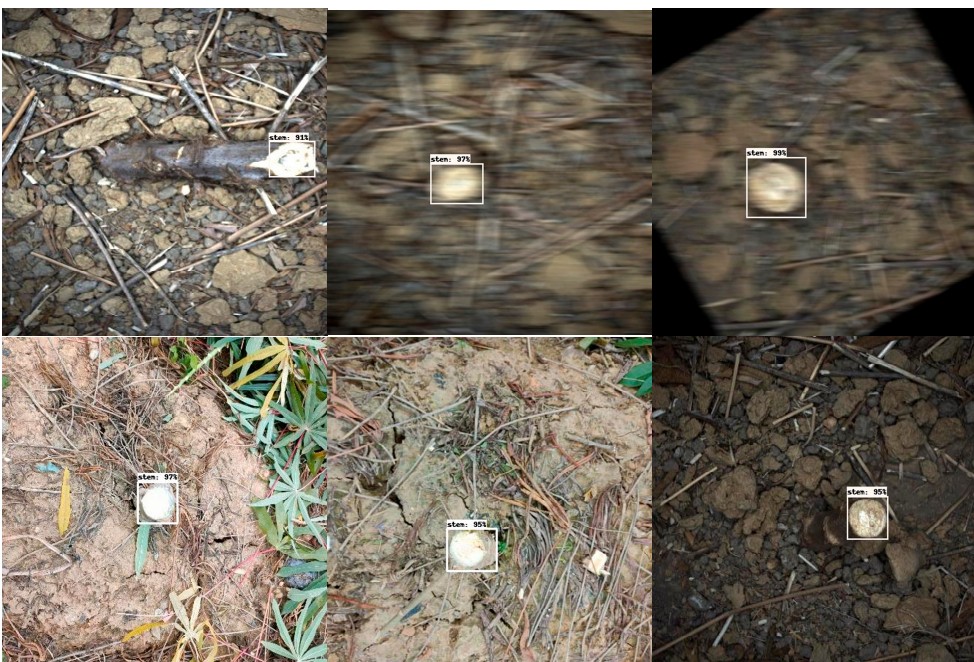

**Figure 13.** KMC-YOLO test results.

In general, the KMC-YOLO performed excellently, achieving better detection of cassava stalks under different conditions during the network test.

Results and Analysis of Ablation Studies and Comparison Tests

The upper section of Table 6 displays the results of the ablation studies, while the lower section displays the results of the comparison tests. According to the ablation studies, the size of the KM1-YOLO model is significantly reduced after replacing the backbone network with MobileNetV1, compared to the YOLOv4 model after K-means clustering (K-YOLO). This demonstrates that replacing the YOLOv4 backbone structure with the MobileNet network can achieve network lightweight, but the accuracy reduces by 4.4 percentage points. Compared to the prior model, the accuracy of the network model has been enhanced after being replaced with MobileNetV2, although the model size has more than doubled. The accuracy of the MobilnetV3 model with the self-contained SE attention module improves with small model size growth, indicating that the adoption of the attention module can improve model accuracy with little model size growth. The use of the MobilnetV3-small model with its own SE module sacrifices detection accuracy, even though the model size can be reduced relative to KM3-YOLO. Despite being able to employ a smaller model than KM3-YOLO, the MobilnetV3-small model with its own SE module reduces detection accuracy. In comparison to the other sets of models, the KMC-YOLO model suggested in this study, which combines the MobileNetV2 model and CA module, has a bigger accuracy improvement as well as improved model size and detection speed.

From the comparison test, it can be seen that the network detection performance of KMC-YOLO is better than the YOLOv3-SPP network and Faster R-CNN network in terms of detection accuracy and detection speed. Its detection accuracy is substantially higher than that of the YOLOv4-tiny, YOLOX-tiny, and YOLOv5s, although its detection speed is not as fast. The harvester clamping and pulling mechanism requires a high stalk–field detection accuracy to offer accurate stalk location and a short detection speed to allow the machine to make changes. The current model has excellent detection accuracy as well as detection speed to match the time requirements of the machinery, allowing it to be used in field detection.

**Table 6.** Comparison of network detection performance.

| Network Model | Backbone | Precision/% | Model Size/M | Detection Speed under GPU/fps |
|---|---|---|---|---|
| K-YOLO | CSPDarknet53 | 95.6 | 245.3 | 19.23 |
| KM1-YOLO | MobileNetV1 | 91.2 | 51.1 | 30.7 |
| KM2-YOLO | MobileNetV2 | 93.1 | 113.2 | 33.4 |
| KM3-YOLO | MobileNetV3 | 94.8 | 114.2 | 28.5 |
| KM3-YOLO (small) | MobileNetV3-small | 94.0 | 110.6 | 34.5 |
| **KMC-YOLO** | **MobileNetV2 + CA** | **98.2** | **115.1** | **33.6** |
| YOLOv3-SPP | Darknet53 | 86.2 | 71.6 | 25.3 |
| YOLOv4-tiny | CSPDarknet53-tiny | 71.4 | 22.4 | 48.5 |
| YOLOX-tiny | CSPDarknet53-tiny | 86.43 | 19.4 | 48.6 |
| YOLOv5s | CSP + Focus | 89.5 | 14.8 | 52.4 |
| Faster R-CNN | ResNet50 + FPN | 84.6 | 137 | 18.5 |

*3.2. Field Validation Trials*

The detection speed of the KMC-YOLO network deployed directly into the Xavier development board is 21.4 fps with 98.1% precision. After being accelerated by TensorRT with FP16 floating-point numbers, the detection speed of KMC-YOLO is 39.3 fps with 96.8% precision, meeting the requirements for field detection. Real-time detection experiments were conducted on cassava stalk sections in the field to verify the model's landing performance after the development of the detection interface was completed. After truncation of cassava stalks in the field, the cross-section of the stalks undergo oxidization and blackening within a certain period of time, which cannot support long-term and repeated tests, so a video stream shot in cassava planting area B of Guangxi University's College of Agriculture, a teaching and research base, was used for the test. The video was shot on 11 January 2022, and the field environment information is shown in Table 7. The cassava stalks were truncated before shooting.

**Table 7.** Field environment information.

| Weather | Environmental Temperature | Relative Humidity | Average Surface Soil Moisture Content | Cassava Variety |
|---|---|---|---|---|
| Sunny | 8–15 °C | 71% | 17.22% | GR891, Bread Cassava No.1 |

Three different time periods were chosen for shooting: 6:30–7:00 a.m., 11:00–12:00 p.m., and 6:00–6:30 p.m. The illumination value during noon was above 10,000 lux, while the illumination value during the early morning and evening was around 1000 lux.

The shooting equipment was fixed on the cassava harvester, and the cassava harvester moved in the same way as during normal operation. The videos captured during the three time periods were saved separately. Videos of the detection process were recorded at three different times and three frames were extracted from each video. Figure 14 displays the extracted images during the test, and all three time periods yielded good recognition results. The overall detection success rate exceeded 95% in all three cases, as presented in Table 8. Upon analysis, false detections were observed in cases of strong light reflection during high noon when illumination values were high. Conversely, missed detections occurred during the morning and evening when illumination values were low. In the field test, the high illuminance value at noon was identical to the simulated environment of the indoor field image dynamic simulation acquisition platform, resulting in robust identification results and even misclassifying clods with high light backlighting as cassava stalks. The misdetection was caused by the low illuminance value in the morning and evening, as well as the gap between natural light and stabilized incandescent light in the interior platform.

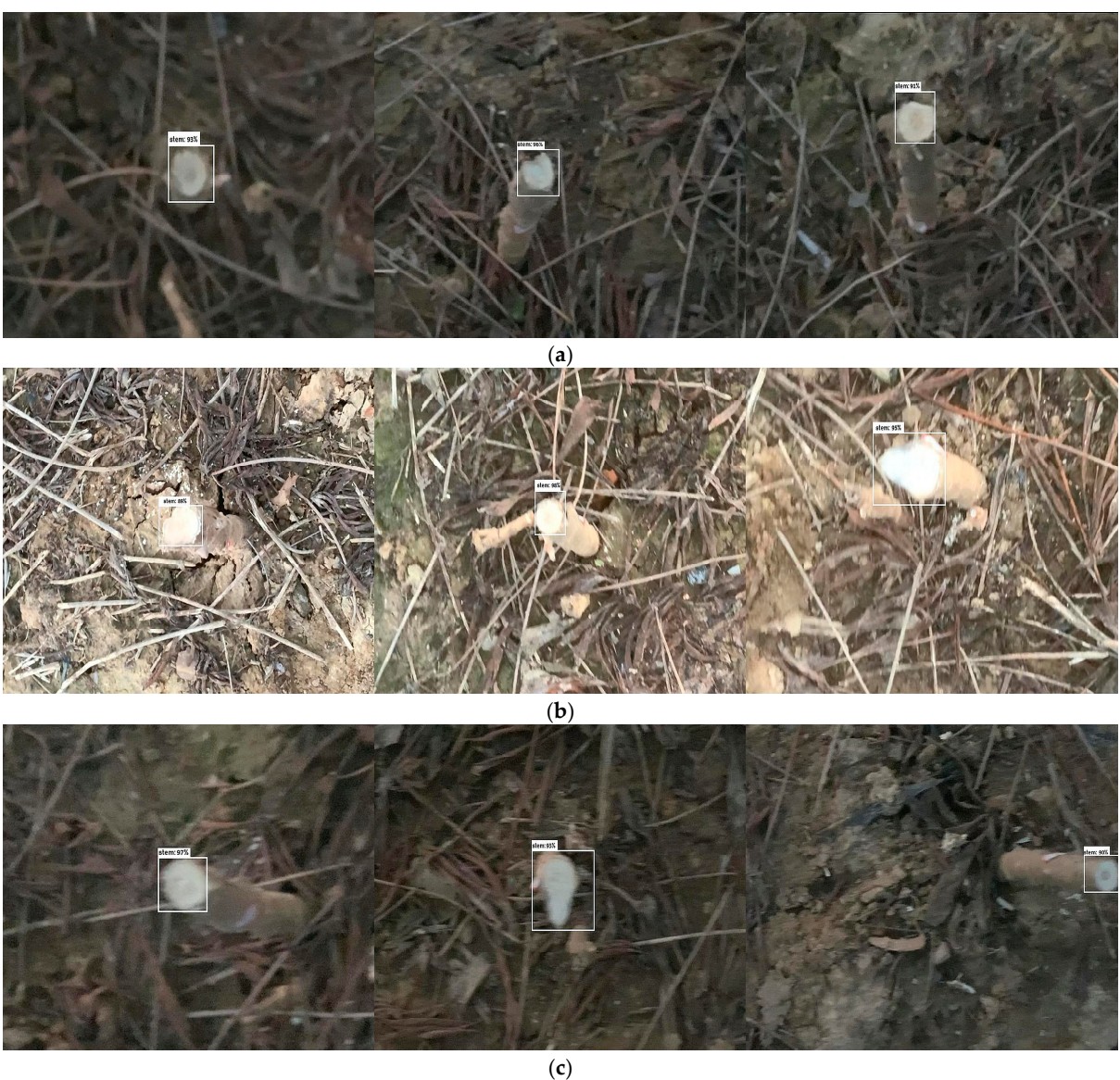

**Figure 14.** Images of the testing process: (**a**) low light detection screen in the morning; (**b**) high light detection screen at noon; (**c**) low light detection screen in the evening.

**Table 8.** Detection statistics.

| Detection Time | Number of Stalks Tested | Number of Correct Detections | Number of Non-Detects | Number of Error Detection | Success Rate/% |
|---|---|---|---|---|---|
| Morning | 268 | 257 | 11 | 0 | 95.8 |
| Noon | 306 | 295 | 0 | 11 | 96.4 |
| Evening | 253 | 241 | 12 | 0 | 95.2 |

## 4. Conclusions

In this paper, a real-time cassava stalk localization detection method was proposed to adjust the movement of the clamping–pulling mechanism. The accuracy and real-time performance of the improved network were verified by comparing it with different models and conducting adaptive improvement. We also evaluated the effectiveness of the model after deploying it on an embedded device, NVIDIA Jetson AGX Xavier, through field trial validation. The following conclusions can be drawn from the comparison, evaluation, and validation:

(1) For the lightweight design of the YOLOv4 model, the KMC-YOLO network was constructed by incorporating the MobileNetV2 + CA module. The AP of the model was tested to be 98.2%, with detection speeds of 33.6 fps and model size reductions of 53.08%. The KMC-YOLO network is suitable for deployment on the Xavier development board.

(2) By deploying the KMC-YOLO network on the NVIDIA Jetson AGX Xavier with TensorRT acceleration, the detection speed of the network on the development board increased to 39.3 fps, which is 83.64% higher than the non-accelerated speed, satisfying the requirement of real-time detection in the field.

(3) The field test validation under different illumination conditions shows that the detection success rate of the model was above 95% under all illumination values tested, demonstrating that the algorithm met the detection requirements of the digging–pulling cassava harvester.

The detection method proposed in this paper meets the requirements for real-time and accurate detection in field harvesting and has significant advantages in terms of small memory usage and fast model operation. The proposed improved model can provide important technical support for the development of digging–pulling cassava harvesters and intelligent harvesting research. However, with the optimization of the algorithm, there is still room for further optimization of the model detection efficiency while maintaining a good detection effect. To accommodate the application of the target detection algorithm in this paper, we collected cassava stalk diameters and deflection angles via extensive statistical field trials, which guided the design of the clamping–pulling mechanism and its opening size. Now, we have manufactured a new digging–pulling cassava harvester. Finally, a cassava tuber harvesting test will be conducted in the field to completely check the operation of the cassava harvesting machine equipped with the upper vision device.

**Author Contributions:** Conceptualization, W.Y., J.X., Z.W. and Z.L.; methodology, J.X. and Z.W.; software, J.X. and Z.W.; validation, D.Z. and Y.H.; formal analysis, J.X. and Z.W.; investigation, D.Z. and Y.H.; resources, W.Y.; data curation, Z.W.; writing—original draft preparation, J.X.; writing—review and editing, W.Y., Z.L. and X.Z.; visualization, J.X.; supervision, X.Z.; project administration, W.Y. and Z.L.; funding acquisition, W.Y. All authors have read and agreed to the published version of the manuscript.

**Funding:** This research was funded by the National Natural Science Foundation of China (Grant Nos. 32160422 and 51365005) and the Guangxi Natural Science Foundation (Grant No.2023GXNSFAA026376).

**Institutional Review Board Statement:** Not applicable.

**Data Availability Statement:** The data presented in this study are available in the main manuscript.

**Conflicts of Interest:** The authors declare no conflict of interest.

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
