# Peer review of "Embedded Field Stalk Detection Algorithm for Digging–Pulling Cassava Harvester Intelligent Clamping and Pulling Device"

_agriculture, doi:10.3390/agriculture13112144_

Round 1

Reviewer 1 Report

Comments and Suggestions for Authors

One of the most famous object identification algorithms is YOLO. The latest version of this series, YOLOv5, is perfect for real-time applications because it can process images at up to 1000 frames per second on a single GPU. Thus, it is believed that YOLOv5 is the most accurate and fastest currently available object identification model - in this case, cassava stems. So why do authors use an older version with low quality parameters? They mention him in line 371 and table 6.

There is little information about the cassava harvester used, which uses computer image analysis. It must be remembered that for reliability reasons, adding a new system to a technical facility reduces it significantly.
It is a good practice to include the Latin name of the plant in the title, specifying the variety.

Accelerated durability and reliability tests are required to determine the suitability of this solution for agricultural practice.

Comments on the Quality of English Language

ok

Author Response

Thank you for your kind letter of “Manuscript ID: agriculture-2682415” on October 18, 2023. Those comments are all valuable and very helpful for revising and improving our paper. We have studied comments carefully and have made correction which we hope meet with approval. Revised portion are highlighted in yellow in the paper.

Here below is our description on revision according to the reviewers’ comments. 

To reviewer 1:

  1. The reviewer’s comment:Why do authors use an older version with low quality parameters?

The authors’ Answer: As experts say, the YOLO series is the most famous object recognition algorithm, and many scientists have worked on it. YOLOv4 and YOLOv5 were released within two months of each other, while YOLOv4 was more customizable, and at that time, most researchers chose YOLOv4 to make improvements and got better results. Adding an intelligent recognition system to a cassava harvester is a new endeavor that requires a reliable recognition network, so we chose YOLOv4 for adaptation. After improvement, the current algorithm has the speed and accuracy that can meet the needs of using and satisfy the current machine, so we think this improvement method and overall program are feasible.

After time verification, YOLOv5 is a mature object detection scheme. Therefore, we also tested it for network performance, and the results are as shown in Table 6 in row 354. Your opinion is in line with our understanding, so we will make adaptations to the new algorithm using similar improvement ideas as YOLOv4, with a view to its application in the new cassava harvester.

  1. The reviewer’s comment: It is a good practice to include the Latin name of the plant in the title, specifying the variety.

The authors’ Answer: Thanks to your suggestion. We have indicated the Latin name of cassava in the abstract and introduction and the Latin name can indicate the broad class of cassava varieties. We think cassava harvester is an industry-specific name that does not require further labeling with the Latin name. Therefore, it has not been changed in the title. The changes in the text are shown below:

Abstract: Cassava (Manihot esculenta crantz) is a major tuber crop worldwide, but its mechanized harvesting is inefficient.

Introduction: Cassava (Manihot esculenta crantz) is one of the world's three major crops, the third largest food crop in the tropical region, the world's sixth largest food crop, 1 billion people's food rations, known as "the king of starch".

  1. The reviewer’s comment:There is little information about the cassava harvester used, which uses computer image analysis. It must be remembered that for reliability reasons, adding a new system to a technical facility reduces it significantly. Accelerated durability and reliability tests are required to determine the suitability of this solution for agricultural practice.

The authors’ Answer: Yes, information on the use of visual identification systems on cassava harvesters is still almost non-existent, while the addition of new systems in technical facilities can affect the reliability of the facilities. Thank you very much for your reminder. The purpose of this article is to investigate the practice of combining cassava mechanical harvesting with visual recognition in order to prepare for the following use of intelligent digging-pulling cassava harvester. Additionally, we have constructed a prototype machine, and the next stage will be to carry out field trials of cassava harvesting in order to fully test how well these machines work. This section has been added at the end. The additions are shown below:

To accommodate the application of the target detection algorithm in this paper, we collected cassava stalk diameters and deflection angles rang by extensive statistical field trials, which guided the design of the clamping-pulling mechanism and its opening size. Now, we have manufactured a new digging-pulling cassava harvester. Finally, a cassava tuber harvesting test will be conducted in the field to completely check the operation of the cassava harvesting machine equipped with the upper vision device.

Reviewer 2 Report

Comments and Suggestions for Authors

The article proposed a field cassava stem detection method based on an improved YOLOv4 model. This method was implemented through an embedded system, and experimental results demonstrated its promising performance. However, there is still a need for further exploration regarding how to effectively integrate this method with the mechanical structure. My comments are listed below:

1. The description of the workflow of the digging-pulling cassava harvester was provided in lines 64-67, while the state of cassava stems during harvesting was described in lines 161-163. This arrangement makes the description somewhat unclear. In my opinion, the description of the harvesting scenario should precede the workflow of the harvester.

2. Lines 167-181 described the process of image acquisition. However, it is not clear what triggers the image acquisition during the harvester's operation. Is it based on human judgment and manual capturing? Provide an explanation regarding the triggering conditions and methodology of image acquisition during the harvester's operation.

3. As shown in Figure 14, when detecting the top of cassava stems using a single camera, I think the thickness and orientation of the cassava stems could influence the position of the cassava stem in the image. If the adjustment of the clamping mechanism is based on the detected position of the cassava stem’s top, how was this consideration taken into account? I recommend adding a discussion on this aspect in the manuscript.

4. The labeling in Figure 1(c) is unclear and requires modification.

Comments on the Quality of English Language

The writing quality basically meets the standards of academic journals, please make minor modifications.

Author Response

Thank you for your kind letter of “Manuscript ID: agriculture-2682415” on October 18, 2023. Those comments are all valuable and very helpful for revising and improving our paper. We have studied comments carefully and have made correction which we hope meet with approval. Revised portion are highlighted in yellow in the paper.

Here below is our description on revision according to the reviewers’ comments. 

To reviewer 2:

  1. The reviewer’s comment:The description of the workflow of the digging-pulling cassava harvester was provided in lines 64-67, while the state of cassava stems during harvesting was described in lines 161-163. This arrangement makes the description somewhat unclear. In my opinion, the description of the harvesting scenario should precede the workflow of the harvester.

The authors’ Answer: Your suggestion is very valuable. The digging-pulling cassava harvester workflow in rows 64-67 is a conventional mechanical workflow, and to avoid misunderstanding, we identify the conventional operation workflow. After describing the cassava stalks at the acquisition site, we describe the workflow of the digging-pulling cassava harvester equipped with an embedded inspection platform. The additions are shown below:

The working principle of a digging-pulling cassava harvester equipped with an embedded inspection platform is as follows: when the cassava harvester travels along the ridge in a straight line, the camera vertically arranged at the front end of the cassava harvester frame acquires the positional information about the cross-section of the cassava stalks on the ridge in the field of view, and transmits the information to the slave computer (Programmable Logic Controller) after processing. The slave computer controls the movement of the clamping-pulling mechanism according to the information, and finally the clamping-pulling mechanism accurately clamps the cassava stalks and completes the pulling and harvesting action.

  1. The reviewer’s comment: Lines 167-181 described the process of image acquisition. However, it is not clear what triggers the image acquisition during the harvester's operation. Is it based on human judgment and manual capturing? Provide an explanation regarding the triggering conditions and methodology of image acquisition during the harvester's operation.

The authors’ Answer: Your suggestion is very meaningful and it was an oversight on our part not to clarify this issue. The description of the problem has now been added to the text, and the additional part is shown below.

During the acquisition process, the cassava harvester simulates the harvesting state by driving along the ridge in a straight line. When the collector observes that the cassava stalk cross-section is fully presented on the screen, the cassava stalk cross-section photographs are collected manually.

  1. The reviewer’s comment:As shown in Figure 14, when detecting the top of cassava stems using a single camera, I think the thickness and orientation of the cassava stems could influence the position of the cassava stem in the image. If the adjustment of the clamping mechanism is based on the detected position of the cassava stem’s top, how was this consideration taken into account? I recommend adding a discussion on this aspect in the manuscript.

The authors’ Answer: Yes, the mechanical structure used to precisely clamp the cassava stalks would be somewhat impacted by the thickness and orientation of the stalks. In order to ensure successful clamping, we conducted numerous statistical tests in the field, which guided the design of the clamping and pulling device and its opening, to determine the relationship between the stalk thickness, the deflection angle, and the clamping device's opening. This section is supplemented at the end. The additions are shown below:

To accommodate the application of the target detection algorithm in this paper, we collected cassava stalk diameters and deflection angles rang by extensive statistical field trials, which guided the design of the clamping-pulling mechanism and its opening size. Now, we have manufactured a new digging-pulling cassava harvester. Finally, a cassava tuber harvesting test will be conducted in the field to completely check the operation of the cassava harvesting machine equipped with the upper vision device.

  1. The reviewer’s comment:The labeling in Figure 1(c) is unclear and requires modification.

The authors’ Answer: Thank you for your suggestion, we have made the change in the text.

Reviewer 3 Report

Comments and Suggestions for Authors

To achieve real-time adjustment of the clamping mechanism of the digging-pulling cassava harvester in the field according to the position of cassava stems, a cassava stem detection method based on improved YOLOv4 was proposed. This method can be implemented through embedded systems, and field experiments have shown that the proposed method exhibits good detection accuracy and real-time performance. The method has the potential to enhance the efficiency of mechanical cassava harvesting in outdoor environments. My comments are listed below:

1. The detection algorithm was implemented using an embedded system, which is tailored for outdoor environments. The use of high-power graphics cards in outdoor settings can be expensive. The necessity of using embedded systems should be emphasized in the Introduction section. I recommend supplementing relevant descriptions.

2. In lines 172-174, the camera was mounted on the cassava harvester frame and was arranged vertically in the field. What are the reasons for choosing this arrangement, and how is it integrated with the adjustment of the clamping mechanism?

3. What is the overall workflow of the harvester after detecting cassava stems in the field? I suggest supplementing relevant descriptions in the discussion section.

4. There are still some grammar errors in the article. I recommend reviewing and revising the sentences for improved accuracy.

Author Response

Thank you for your kind letter of “Manuscript ID: agriculture-2682415” on October 18, 2023. Those comments are all valuable and very helpful for revising and improving our paper. We have studied comments carefully and have made correction which we hope meet with approval. Revised portion are highlighted in yellow in the paper.

Here below is our description on revision according to the reviewers’ comments. 

To reviewer 3:

  1. The reviewer’s comment: The detection algorithm was implemented using an embedded system, which is tailored for outdoor environments. The use of high-power graphics cards in outdoor settings can be expensive. The necessity of using embedded systems should be emphasized in the Introduction section. I recommend supplementing relevant descriptions.

The authors’ Answer: Thank you very much for your suggestion. We have added the necessity of using embedded systems in the field in the introduction section. The additional section is shown below:

Although object detection algorithms are increasingly being utilized in agricultural engineering, the working environment of agricultural machinery in the field is terrible, and the expense of establishing large-scale computing platforms on it is prohibitively expensive, with no guarantee of dependability. Embedded platforms are tiny in size and can give more performance and reduced power consumption, as well as greater dependability and security, and can provide field edge computing at a cheaper cost [26]. On the other hand, the computational complexity of deep learning networks remains a challenge for their practical implementation in vehicular field environments with limited computing power.

  1. Zhang, Y.; Yu, J.; Chen, Y.; Yang, W.; Zhang, W.; He, Y. Real-time strawberry detection using deep neural networks on embedded system (rtsd-net): An edge AI application. Comput Electron Agr2022, 192, 106586. [CrossRef]

  1. The reviewer’s comment:In lines 172-174, the camera was mounted on the cassava harvester frame and was arranged vertically in the field. What are the reasons for choosing this arrangement, and how is it integrated with the adjustment of the clamping mechanism?

The authors’ Answer: Thank you very much for your question, we were not very clear in the paragraph before and have now made the following change in the paragraph.

Specifically, the industrial camera was mounted on the front end of the cassava harvester frame, and the lens was arranged vertically with the field to detect the top section of the cassava stalk, as shown in Figure 6. The camera is arranged before the clamping mechanism, which can reserve enough time for the movement of the clamping-pulling mechanism; the camera is arranged vertically in the field, so that the camera field of view and the lower plane of the frame can be roughly coincident, and a machine coordinate system is constructed in which the camera detection field of view and the frame of the harvester coincide. In this way, the harvester can more easily obtain the position information of the cassava stalks relative to the rack, facilitating the harvester clamping device to directly obtain the position coordinates of the cassava stalks for real-time adjustment, and also reducing the use of arithmetic power.

  1. The reviewer’s comment:What is the overall workflow of the harvester after detecting cassava stems in the field? I suggest supplementing relevant descriptions in the discussion section.

The authors’ Answer: It’s an excellent suggestion to provide an overall harvester workflow in the paper. A description of the overall workflow of the harvester after detecting cassava stalks in the field in lines 192-200 was provided. We believe such revisions are more appropriate for the reader to learn a full-text pulse and make our following work easier to understand.

The working principle of a digging-pulling cassava harvester equipped with an embedded inspection platform is as follows: when the cassava harvester travels along the ridge in a straight line, the camera vertically arranged at the front end of the cassava harvester frame acquires the positional information about the cross-section of the cassava stalks on the ridge in the field of view, and transmits the information to the slave computer (Programmable Logic Controller) after processing. The slave computer controls the movement of the clamping-pulling mechanism according to the information, and finally the clamping-pulling mechanism accurately clamps the cassava stalks and completes the pulling and harvesting action.

  1. The reviewer’s comment:There are still some grammar errors in the article. I recommend reviewing and revising the sentences for improved accuracy.

The authors’ Answer: We have grammar-checked and made changes to the full text.